# Structures of diverse poxin cGAMP nucleases reveal a widespread role for cGAS-STING evasion in host–pathogen conflict

**James B Eaglesham[1,2,3], Kacie L McCarty[1,2], Philip J Kranzusch[1,2,3,4]\***

[1]Department of Microbiology, Harvard Medical School, Boston, United States; [2]Department of Cancer Immunology and Virology, Dana-Farber Cancer Institute, Boston, United States; [3]Harvard PhD Program in Virology, Division of Medical Sciences, Harvard University, Boston, United States; [4]Parker Institute for Cancer Immunotherapy at Dana-Farber Cancer Institute, Boston, United States

**Abstract** DNA viruses in the family *Poxviridae* encode poxin enzymes that degrade the immune second messenger 2′3′-cGAMP to inhibit cGAS-STING immunity in mammalian cells. The closest homologs of poxin exist in the genomes of insect viruses suggesting a key mechanism of cGAS-STING evasion may have evolved outside of mammalian biology. Here we use a biochemical and structural approach to discover a broad family of 369 poxins encoded in diverse viral and animal genomes and define a prominent role for 2′3′-cGAMP cleavage in metazoan host-pathogen conflict. Structures of insect poxins reveal unexpected homology to flavivirus proteases and enable identification of functional self-cleaving poxins in RNA-virus polyproteins. Our data suggest widespread 2′3′-cGAMP signaling in insect antiviral immunity and explain how a family of cGAS-STING evasion enzymes evolved from viral proteases through gain of secondary nuclease activity. Poxin acquisition by poxviruses demonstrates the importance of environmental connections in shaping evolution of mammalian pathogens.

**\*For correspondence:**
philip_kranzusch@dfci.harvard.edu

**Competing interests:** The authors declare that no competing interests exist.

## Introduction

The cGAS-STING pathway is a major sensor of pathogen infection in mammalian cells where it functions to detect mislocalized cytosolic DNA exposed during infection (*Ablasser and Chen, 2019*). The enzyme cyclic GMP–AMP synthase (cGAS) binds directly to cytosolic DNA, and becomes activated to synthesize the nucleotide second messenger 2′–5′/3′–5′ cyclic GMP–AMP (2′3′-cGAMP) (*Sun et al., 2013*). 2′3′-cGAMP is then recognized by the receptor Stimulator of Interferon Genes (STING), which oligomerizes and recruits downstream signaling adapters to drive induction of type I interferon and NF-κB signaling (*Figure 1A*; *Ablasser et al., 2013*; *Diner et al., 2013*; *Gao et al., 2013*; *Zhang et al., 2019*; *Zhang et al., 2013*; *Zhao et al., 2019*).

In order to productively replicate in mammalian cells, viruses must evade immune surveillance. Poxviruses are large DNA viruses which replicate exclusively in the cytosol (*Moss, 2013*), and encode poxvirus immune nucleases (poxins) to degrade 2′3′-cGAMP and prevent STING activation (*Figure 1A*; *Eaglesham et al., 2019*). Vaccinia virus (VACV) poxin (encoded by the gene *B2R*) is sufficient to antagonize cGAS-STING signaling in cells and is necessary for effective viral replication in vivo. Crystal structures of VACV poxin in the pre- and post-reactive states revealed that catalysis proceeds through a metal-independent mechanism, contorting 2′3′-cGAMP into a conformation that activates the 2′ hydroxyl for in-line cleavage of the 3′–5′ bond (*Eaglesham et al., 2019*).

**Figure 1.** Poxin 2'3'-cGAMP nuclease activity is conserved across three families of viruses and two orders of insects. (**A**) Schematic of cGAS-STING signaling. The sensor cyclic GMP–AMP synthase (cGAS) detects cytosolic DNA and synthesizes the second messenger 2'3'-cGAMP from ATP and GTP. 2'3'-cGAMP activates the receptor Stimulator of Interferon Genes (STING) and initiates downstream antiviral signaling. Poxin enzymes inhibit cGAS-STING signaling by degrading 2'3'-cGAMP and blocking activation of STING. (**B**) Bioinformatic identification (*Figure 1—figure supplement 1A*) and biochemical verification of diverse poxin enzymes. Poxins can be divided into groups, including enzymes from mammalian and insect poxviruses (Group 1), parasitoid wasps and alphabaculoviruses (Group 2), moths and butterflies (Lepidoptera) (Group 3), and cypoviruses, betabaculoviruses, and betaentomopoxviruses (Group 4). Poxin 2'3'-cGAMP nuclease activity is conserved within each group. Recombinant proteins (*Figure 1—figure supplement 1B*) were incubated with radioactively labeled 2'3'-cGAMP for 1 hr at 37°C, and degradation products were resolved using thin-layer chromatography (TLC). Four proteins could not be expressed in *E. coli* (red text, empty lanes on TLC plate). See Materials and methods for protein accession numbers. Data are representative of two independent experiments.

The online version of this article includes the following figure supplement(s) for figure 1:

**Figure supplement 1.** Bioinformatic identification and biochemical characterization of poxin homologs.

Poxviruses have a well-documented ability to acquire genes horizontally, especially from their hosts (*Hughes et al., 2010*). The closest homologs of VACV poxin belong not to mammals or other mammalian viruses, but to baculoviruses, and Lepidoptera (moths and butterflies) which serve as exclusive hosts of baculoviruses (*Eaglesham et al., 2019*). While baculovirus and lepidopteran poxin homologs share <25% identity with VACV poxin, they are functional nucleases and retain 2'3'-cGAMP-specific cleavage activity (*Eaglesham et al., 2019*). The distribution of poxin homologs in

mammalian poxviruses, insects, and insect viruses indicates that poxviruses may have obtained this gene through horizontal transfer from an ancestral host-pathogen conflict.

Here we use a forward biochemical approach to map the evolutionary origin of poxin enzymes and define a genetic route through which poxviruses acquired a new mechanism of immune evasion. We determine four X-ray crystal structures of baculovirus and lepidopteran host poxins, revealing an unexpected origin of poxin enzymes as descendants of viral proteases. The structures of the cabbage looper moth *Trichoplusia ni*, and monarch butterfly *Danaus plexippus* poxin enzymes resemble self-cleaving proteases from positive-sense single-stranded RNA ((+)ssRNA) viruses, and bind their own C-termini within a vestigial protease active-site pocket. Using the lepidopteran poxin structures as a guide, we identify functional poxin enzymes in the genomes of unclassified insect-specific RNA viruses distantly related to flaviviruses, which possess both 2′3′-cGAMP nuclease activity and auto-proteolytic cleavage activity. Our results define broad importance for 2′3′-cGAMP cleavage in metazoan host-pathogen conflict and reveal an evolutionary path through which an insect RNA viral protease developed secondary nuclease activity to inhibit cGAS-STING immunity. Conservation of poxin cGAS-STING evasion among pathogens like monkeypox virus and cowpox virus highlights a deep genetic connection which allowed these mammalian poxviruses to obtain a new mechanism of immune evasion from the environment.

## Results

### Poxins are a diverse family of 2′3′-cGAMP nucleases

To define poxin diversity and phylogenetic distribution, we used both the VACV poxin and lepidopteran *Trichoplusia ni* poxin sequences to seed position-specific iterative BLAST (PSI-BLAST) searches, identifying an initial combined total of 351 unique poxin-like sequences. Poxin homologs can be classified into four major enzyme groups with <25% identity to one another and which share seven different phylogenetic origins (*Figure 1B*, *Figure 1—figure supplement 1A*). We cloned 33 diverse representatives and directly tested recombinant protein for 2′3′-cGAMP nuclease activity using thin-layer chromatography (*Figure 1B*). Poxin homologs from all groups efficiently degraded 2′3′-cGAMP (*Figure 1B*), demonstrating nuclease activity is a conserved function of this protein family regardless of the sequence origin or genomic context.

The presence of active poxin homologs within each major group demonstrates widespread distribution of an enzyme family dedicated to regulation and evasion of cGAS-STING signaling. Poxins are encoded in a diverse array of viruses and host animal species (*Figure 1B*, *Supplementary file 1*). Group one is composed of enzymes from mammalian and insect poxviruses, some of which are fused to a C-terminal schlafen domain (*Eaglesham et al., 2019*; *Liu et al., 2018a*). Groups 2–4 consist entirely of enzymes identified in the genomes of moths and butterflies (Lepidoptera), and viruses or parasites which infect these insects. Group two primarily contains poxins encoded in alphabaculovirus genomes. However, two sequences identified in the genomes of parasitoid wasps (*Microplitis demolitor* and *Glypta fumiferanae*) cluster alongside these viral enzymes. These wasps parasitize caterpillars, laying their eggs inside them and co-injecting domesticated bracoviruses which modulate caterpillar immunity to favor egg maturation (*Béliveau et al., 2015*; *Burke et al., 2018*; *Strand and Burke, 2015*). Lepidopteran poxin enzymes cluster within Group 3, and have been shown to be highly upregulated after infection with various pathogens, including alphabaculoviruses and bacteria, providing evidence for a role in immunity or immune regulation (*Shrestha et al., 2019*; *Woon Shin et al., 1998*). Group four is formed by poxin proteins from the genomes of several additional lepidopteran viruses: betabaculoviruses (also called granuloviruses), cypoviruses (double-stranded RNA viruses in the family *Reoviridae*), and betaentomopoxviruses (*Figure 1B*; *Silva et al., 2020*). The vast evolutionary distance between these DNA and RNA viral families suggests related poxin genes may have been transferred between viruses within coinfected cells (*Silva et al., 2020*; *Thézé et al., 2015*). Notably, the enormous diversity of poxin enzymes in insect pathogens and moth and butterfly genomes confirms a broad role for 2′3′-cGAMP degradation, and strongly suggests these genomes served as a source for emergence of poxins in mammalian poxviruses.

eLife Research article

Microbiology and Infectious Disease | Structural Biology and Molecular Biophysics

# Host and viral poxins employ alternative catalytic residues for 2′3′-cGAMP cleavage

Poxin enzymes from different groups share at most 15–25% identity, preventing identification of active-site residues and limiting analysis of the mechanism of 2′3′-cGAMP cleavage. To enable comparative analysis with VACV poxin (Group 1), we next determined a series of structures of representative enzymes from Groups 2–4 in complex with 2′3′-cGAMP. New poxin structures include the alphabaculovirus *Autographa californica* nucleopolyhedrovirus (AcNPV) poxin from Group 2, lepidopteran host poxins from the moth *Trichoplusia ni* and the monarch butterfly *Danaus plexippus* in Group 3, and the betabaculovirus *Pieris rapae* granulovirus (PrGV) poxin from Group 4 (*Figure 2A*, *Supplementary file 2*). In spite of dramatic sequence divergence, poxin structures from each group reveal a shared core fold and head-to-tail dimeric architecture confirming poxins as a single enzyme family (*Figure 2A*, *Figure 2—figure supplement 1A*).

The new poxin structures trap a post-reaction state following 2′3′-cGAMP cleavage and allow direct comparison with the catalytic mechanism of VACV poxin (*Eaglesham et al., 2019*). VACV poxin functions through a metal-independent mechanism, contorting 2′3′-cGAMP to position the 2′ hydroxyl for in-line attack on the 3′–5′ bond, generating a cyclic phosphate intermediate which is further resolved into a 3′ phosphate product (*Eaglesham et al., 2019*). Strikingly, despite sharing <21% identity, each poxin structure contorts 2′3′-cGAMP in an identical strained conformation (*Figure 2B–F*). Consistent with specificity for 2′3′-cGAMP and a shared mechanism of catalysis, the active-site charge landscape is conserved across all poxin proteins, with hydrophobic pockets for the bases, and multiple basic residues stabilizing the negatively charged phosphates of the 2′3′-cGAMP ligand (*Figure 2G*, *Figure 2—figure supplement 1C*).

In contrast to the shared conformation of 2′3′-cGAMP during cleavage, poxin enzymes exhibit diverse catalytic amino-acids that activate the 2′ hydroxyl for in-line attack and stabilize cleavage intermediates. Unlike the catalytic triad of histidine, tyrosine, and lysine residues essential for VACV and AcNPV poxin cleavage activity (*Figure 2B,C*, *Figure 2—figure supplement 1B,C*), the active sites of *T. ni* poxin and PrGV poxin are divergent. The *T. ni* and PrGV poxin enzymes share the conserved active-site histidine but lack a tyrosine residue entirely (*Figure 2D,E*). Instead, both proteins contain a second histidine residue adjacent to the first, which forms a contact with the 2′ hydroxyl or 3′ phosphate of the cleaved 2′3′-cGAMP molecule (*Figure 2D,E*, *Figure 2—figure supplement 1C*). In the case of *T. ni* poxin, mutagenesis analysis demonstrates this second histidine residue is essential for activity, suggesting this residue functions as a general base to deprotonate and activate the 2′ hydroxyl nucleophile for in-line cleavage of the 3′–5′ bond (*Figure 2—figure supplement 1B*). Further, the *T. ni* poxin protein shows substitution of the VACV lysine for an arginine residue, whereas the lysine is conserved in PrGV poxin (*Figure 2D,E*). Unlike mutation of the lysine residue in the VACV and AcNPV active sites, mutation of the *T. ni* poxin active-site arginine to alanine only partially reduces activity, and a charge-preserving mutation of the arginine to lysine results in even less activity (*Figure 2—figure supplement 1B*). Together, these results reveal amino-acid diversity in the active-site of poxin enzymes and suggest functional differences in the ability to control 2′3′-cGAMP stability.

To study potential functional variation between viral and host poxins, we measured kinetics for VACV, AcNPV, and *T. ni* poxin 2′3′-cGAMP degradation. (*Figure 2—figure supplement 2A–D*). Interestingly, mammalian and insect viral poxins from VACV and AcNPV exhibit similar kinetics, with a $K_M$ of 0.83 µM and 2.43 µM respectively, while the *T. ni* host poxin protein has a $K_M$ two orders of magnitude higher at 342.1 µM. However, the host *T. ni* poxin has a much higher rate constant of 2254 $min^{-1}$ compared to VACV and AcNPV poxin (93.4 $min^{-1}$ and 346 $min^{-1}$, respectively). For comparison, we determined the $K_D$ of VACV, AcNPV, and *T. ni* poxin for 2′3′-cGAMP using enzymes with an inactivating catalytic site histidine mutation (*Figure 2—figure supplement 2E–H*). Consistent with our enzyme kinetics results, inactive VACV and AcNPV poxin exhibited a $K_D$ of 0.58 µM and 0.81 µM respectively, within a similar range as the respective $K_M$ values of wild-type enzymes. In contrast, we could not measure the $K_D$ of the catalytic site mutant *T. ni* poxin, indicating it is higher than 20 µM, again consistent with the high $K_M$ value for wildtype *T. ni* poxin. These enzymes represent only three members of a highly divergent enzyme family, and additional analyses will be required to determine whether these trends in enzyme kinetics hold true for other viral and host poxin enzymes.

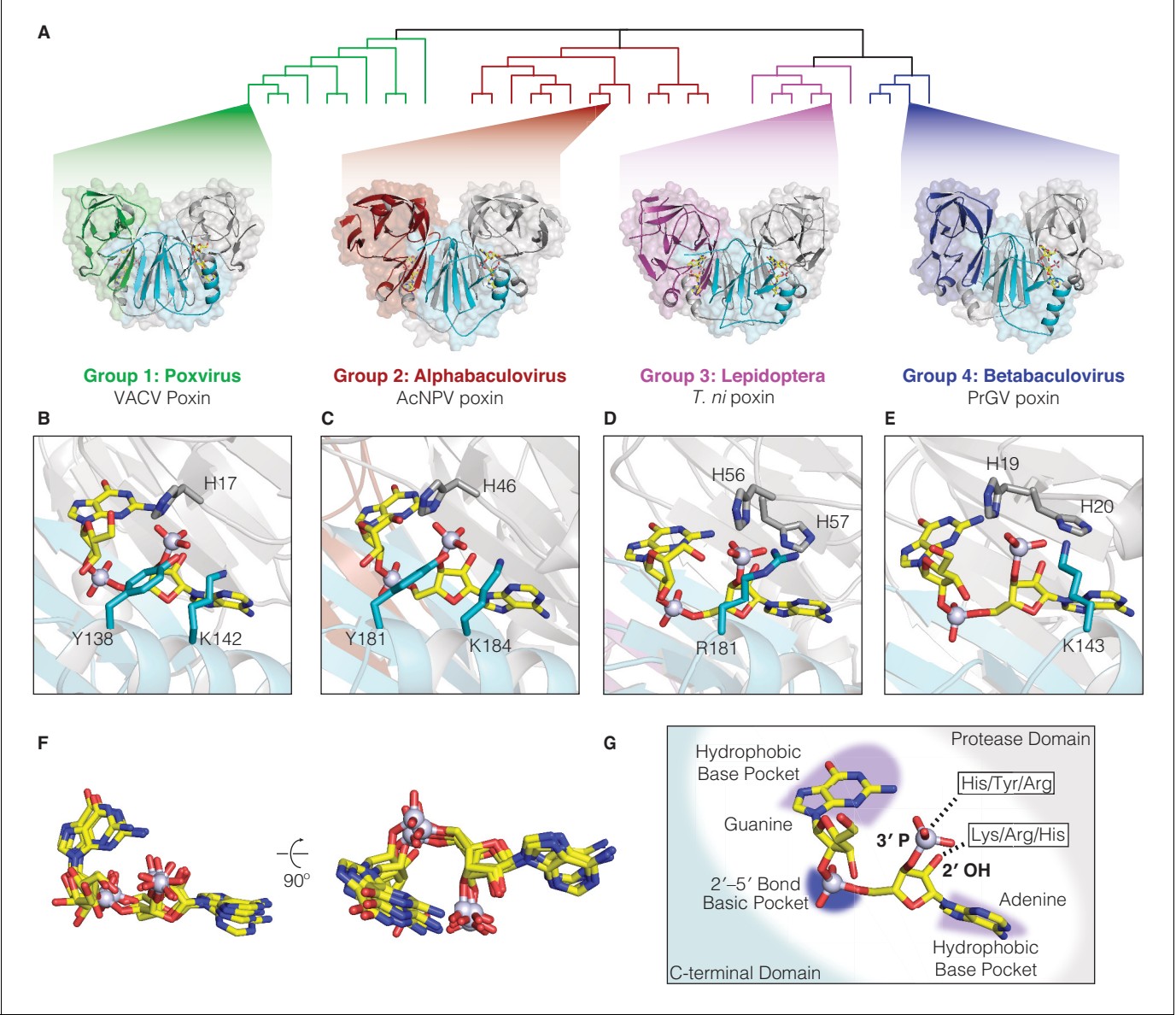

**Figure 2.** Poxin crystal structures define a conserved catalytic mechanism and species-specific active-site adaptations. (A) X-ray crystal structures of poxin enzymes from an alphabaculovirus (AcNPV), the moth *Trichoplusia ni*, and a betabaculovirus (PrGV) allow comparison with VACV poxin (PDB: 6EA9) and demonstrate family-wide structural conservation (*Figure 2—figure supplement 1A*). (B, C) Conservation of the active-site triad between VACV and AcNPV poxin. (D, E) The insect *T. ni* and betabaculovirus PrGV poxin proteins contain altered active-site triads composed of two histidines, and an arginine or lysine residue. Consistent with alternative active-site residues, lepidopteran poxin enzymes show remarkably different kinetic properties (*Figure 2—figure supplement 2*). (F) Despite differences in active-site residues (*Figure 2—figure supplement 1B,C*), 2'3'-cGAMP is contorted within each structure into a similar strained conformation, confirming that poxins operate by the same core metal-independent mechanism established for VACV poxin. (G) Shared poxin active-site features (see *Figure 2—figure supplement 1C*) include hydrophobic pockets for each base of 2'3'-cGAMP and charged interactions that read out the 2'–5' phosphodiester linkage.

The online version of this article includes the following source data and figure supplement(s) for figure 2:

**Figure supplement 1.** Comparison of 2'3'-cGAMP recognition and degradation in insect virus and host poxin enzymes.
**Figure supplement 2.** Host and viral poxins have alternative enzyme kinetics and ligand binding properties.
**Figure supplement 2—source data 1.** Enzyme Kinetics Source Data.
**Figure supplement 2—source data 2.** Enzyme Binding Source Data.

Although animal studies will be required to understand the biological role of diverse poxins, our biochemical analysis is consistent with a model where viral poxins are adapted to depletion of 2′3′-cGAMP to low levels, while the host poxin is adapted for regulation and efficient clearance of 2′3′-cGAMP after accumulation to higher concentrations. Further supporting these results, studies of the lepidopteran transcriptional response to baculovirus infection show that host poxin genes are strongly upregulated during infection in a second wave of transcription occurring after initial immune induction (*Shrestha et al., 2019*; *Woon Shin et al., 1998*). Together, these results reveal that while all poxin enzymes use the same overall mechanism, amino-acid diversity in the active site of poxin enzymes likely enables 2′3′-cGAMP nuclease activity to be tailored to alternative immuno-regulatory and immuno-evasion functions (*Figure 2—figure supplement 2*).

## Structural analysis reveals poxins are descended from self-cleaving RNA-virus proteases

To define the origin of poxin enzymes, we next compared each nuclease against other structures in the Protein Data Bank to identify proteins with related folds. Although poxin functions as a 2′3′-cGAMP-specific nuclease, previous analysis of VACV poxin demonstrated that no structural homology exists with other nuclease or phosphodiesterase enzymes (*Eaglesham et al., 2019*). Instead, the N-terminal domain of VACV poxin exhibits weak homology to chymotrypsin-like serine proteases (*Figure 3A*). Analysis of the host *T. ni* poxin structure confirms a relationship with protease enzymes and demonstrates strong homology with proteases derived from (+)ssRNA viruses (*Figure 3A,B*). Unlike the degenerated domain in VACV poxin, comparison of the N-terminal domain of *T. ni* poxin with the yellow fever virus protease demonstrates near complete conservation of a dual Greek key β-barrel fold common to serine protease enzymes (*Figure 3C*). Like a molecular fossil, *T. ni* poxin likely retains extensive ancestral protease homology due to the slow evolutionary rate of insect host genomes compared to the rapid replication and divergence of viral genes (*Duffy et al., 2008*). In addition to the N-terminal protease domain, all poxins share a C-terminal domain required for dimerization and formation of the nuclease active site. Accordingly, this domain is highly conserved with no degeneration occurring between VACV and *T. ni* poxin (*Figure 3C*). Together these results reveal unexpectedly close homology between poxin and protease enzymes and suggest a nuclease dedicated to 2′3′-cGAMP degradation evolved through dimerization and divergence of an ancient viral protease.

Identification of (+)ssRNA viral proteases as the closest structural homologs to poxin indicates a direct evolutionary connection between these groups of enzymes. (+)ssRNA viruses typically encode gene products as a polyprotein that must be proteolytically cleaved to release individual mature peptides (*Lei and Hilgenfeld, 2017*). Additionally, (+)ssRNA viruses often possess accessory proteases which excise themselves from the polyprotein and serve alternative structural or immune antagonist roles (*Lei and Hilgenfeld, 2017*; *Mann and Sanfaçon, 2019*). Given that poxins function as immune antagonists and share structural homology to (+)ssRNA viral proteases, we hypothesized that poxins originated as self-cleaving accessory nucleases within ancient RNA-virus genomes.

To test this hypothesis, we compared the vestigial protease active site of *T. ni* poxin with the chikungunya virus capsid protein that functions as a self-cleaving accessory protease (*Figure 4A*). In the chikungunya virus capsid protease domain, the cleaved C-terminus remains coordinated in the active site by histidine and aspartic acid residues adjacent to the catalytic serine (*Figure 4A,C*; *Page and Di Cera, 2008*; *Sharma et al., 2018*). A nearly identical pocket is conserved on the surface of the *T. ni* poxin protease-like domain, in which the poxin C-terminus is also coordinated by histidine and aspartic acid residues (*Figure 4B,D,G*, *Figure 4—figure supplement 1*). Close examination of the high-resolution 1.45 Å *D. plexippus* poxin structure allows complete analysis of the interactions between the C-terminal peptide and the surface of the protease-like domain (*Figure 4—figure supplement 1A–C*). The final five residues in the C-terminal peptide in lepidopteran poxin proteins are highly conserved (*Figure 4—figure supplement 1D*) and bind within the protease-domain pocket through hydrophobic interactions where each residue resides within a pocket of corresponding size and shape. Authentic proteases recognize their substrates through similar sidechain-specific interactions within pockets adjacent to the active site, demonstrating that lepidopteran poxins retain a protease-like mechanism for recognition and positioning of the C-terminus within a vestigial protease active site (*Figure 4—figure supplement 1B,C*; *Hedstrom, 2002*). Comparison of lepidopteran poxins with viral poxins from AcNPV and VACV reveals greater degeneration in the protease domain

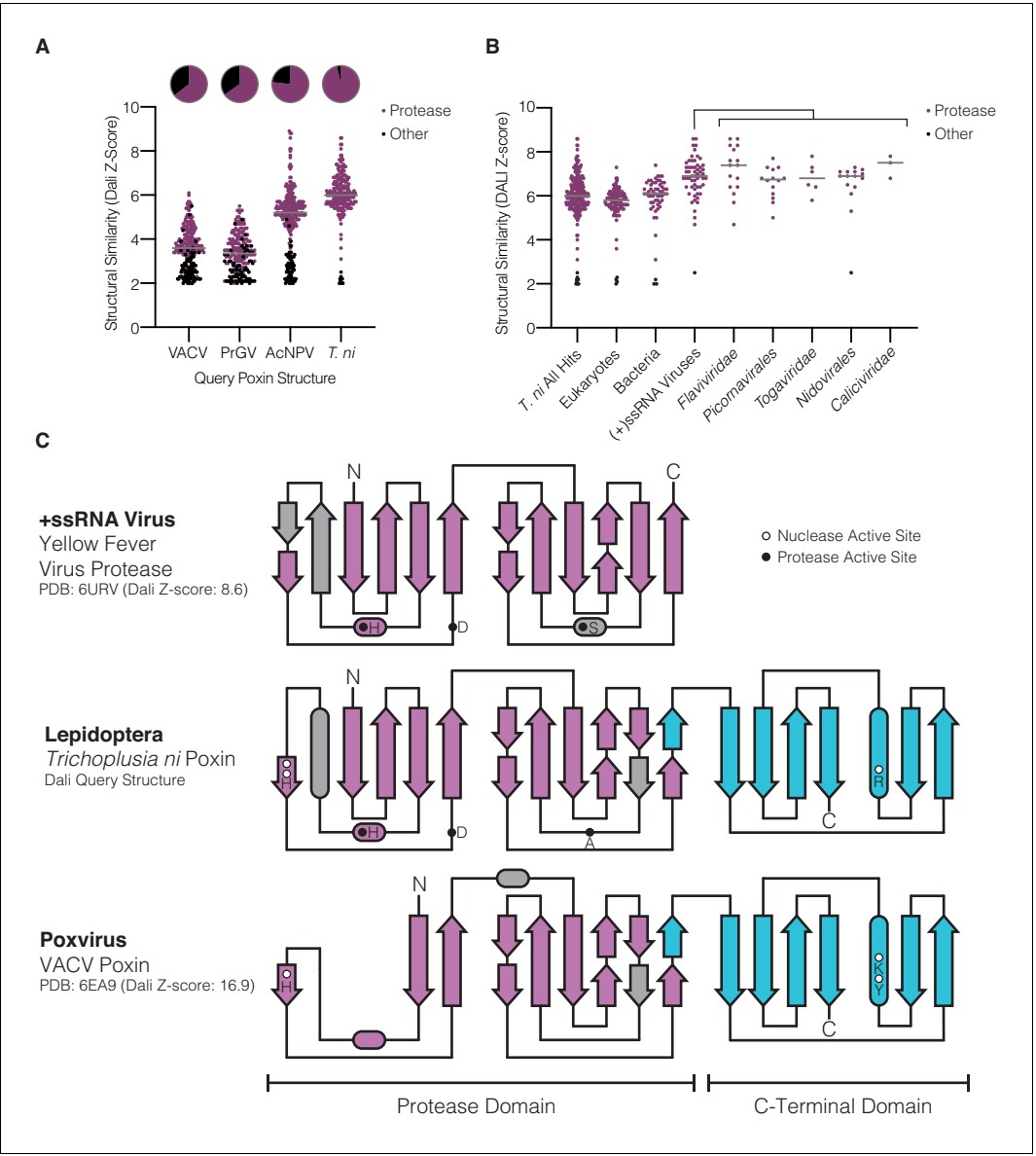

**Figure 3.** Lepidopteran poxins share structural homology with (+)ssRNA viral proteases. (**A**) Dot plot comparing homology of each poxin structure to representatives in the Protein Data Bank. Poxin homologs were identified by DALI and graphed according to Z-score (higher Z-score indicates stronger homology) with a cutoff of 2. Red dots denote homologs that are serine proteases, and the proportion of hits assigned as proteases is indicated in the pie chart above. (**B**) Dot plot as in A depicting homology of *T. ni* poxin to enzymes in the Protein Data Bank, broken down by phylogenetic group. *T. ni* poxin shares the strongest homology with proteases from (+)ssRNA viruses. Gray bars in A and B represent the median Dali Z-score value for each distribution. (**C**) Topology diagram demonstrating conservation of a dual Greek key β-barrel chymotrypsin-like protease fold in the N-terminal domain of *T. ni* poxin, compared with the yellow fever virus protease and VACV poxin. Residues corresponding to the protease active site are denoted as filled circles, and poxin nuclease active-site residues are denoted as open circles.

and protease active-site pocket in viral enzymes that results in loss of C-terminal coordination (*Figure 3A*, *Figure 4E–G*). Notably, the protease active-site pocket is entirely distinct from the poxin nuclease active site, which forms at the dimer interface between poxin monomers (*Figure 2A*, *Figure 4B,G*), explaining how 2'3'-cGAMP nuclease activity could evolve independently within an ancestral self-cleaving accessory protease. Later expression of poxins from a discrete open reading

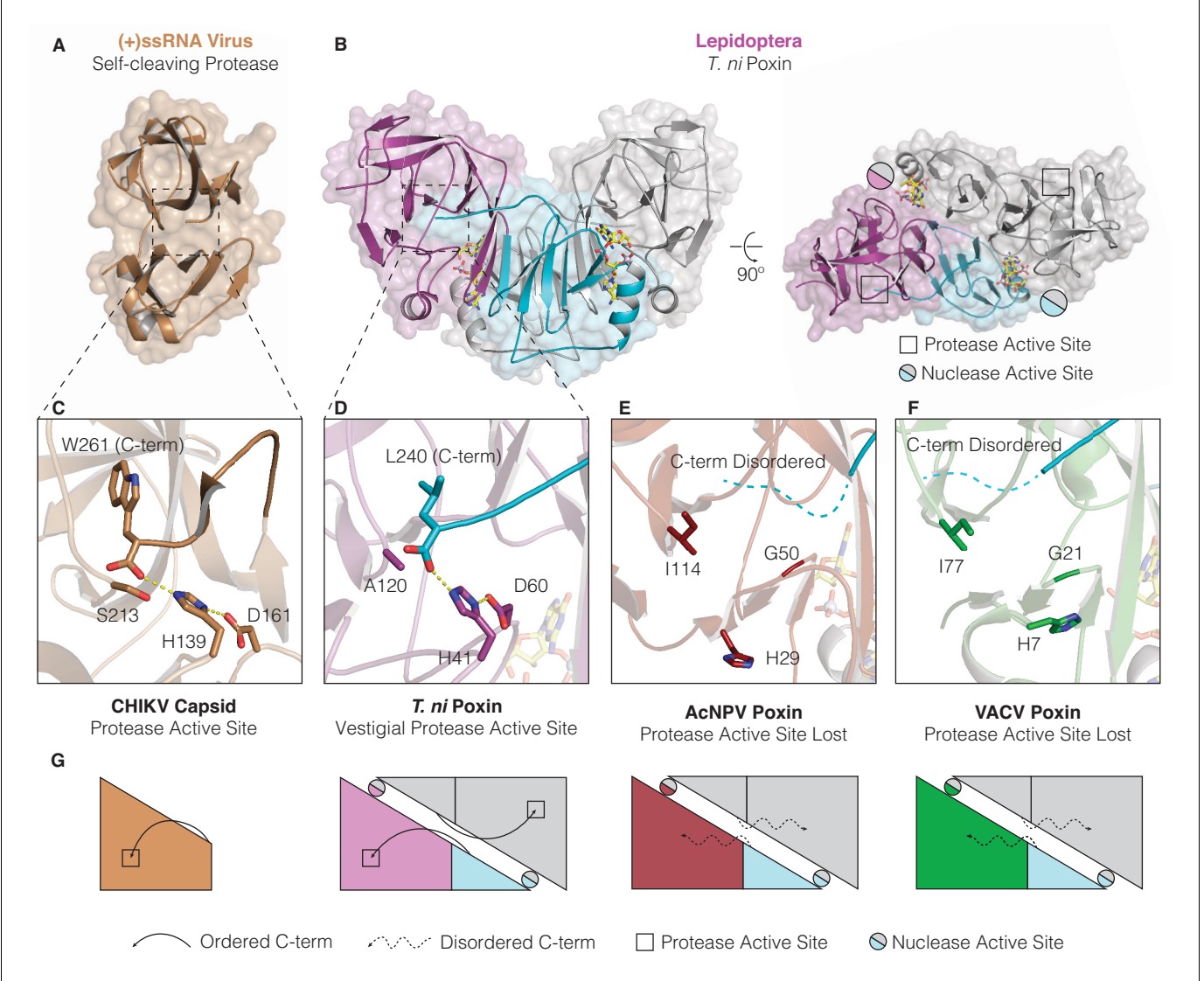

**Figure 4.** Structures of lepidopteran poxins reveal a vestigial self-cleaving protease active site. (**A**) Structure of a (+)ssRNA virus self-cleaving accessory protease, the chikungunya virus capsid protein (PDB: 5H23) (*Sharma et al., 2018*). (**B**) The structure of the lepidopteran *T. ni* poxin reveals a vestigial protease active site (black boxes, at right) that is distinct from the poxin nuclease active site which forms at the dimer interface (circles, at right). Protease-like features are conserved amongst lepidopteran poxins (*Figure 4—figure supplement 1*). (**C, D**) Dotted lines show closeup views of the protease active site of a functional (+)ssRNA virus self-cleaving protease, and the vestigial protease active site of *T. ni* poxin. (**E, F**) Closeups of the same region of AcNPV and VACV poxin as in **D** demonstrating degeneration of the protease active site in viral poxins. (**G**) Schematics indicating how lepidopteran poxins coordinate the C-terminus within a protease active-site pocket similar to the chikungunya virus capsid self-cleaving protease. Although the vestigial protease active site has been lost in AcNPV and VACV poxin, the poxin nuclease active site is preserved at the dimer interface in these enzymes.

The online version of this article includes the following figure supplement(s) for figure 4:

**Figure supplement 1.** Conservation of protease-like features in lepidopteran poxin enzymes.

frame in insect hosts or other viruses likely released selective pressure for maintenance of self-cleavage activity and resulted in protease-domain degeneration.

## Insect (+)ssRNA viruses encode functional self-cleaving poxins

To further test the hypothesis that poxins originated within ancient (+)ssRNA-virus polyproteins we next searched for modern day viral descendants which encode functional self-cleaving poxin enzymes. Re-examination of our initial PSI-BLAST results (*Figure 1—figure supplement 1A*)

revealed additional short (92–195 amino acids), divergent poxin-like regions within large 5,901–8572 amino acid long polyproteins of eight (+)ssRNA flavivirus-like viruses that are unclassified members of the order *Amarillovirales* (*Koonin et al., 2020*). These genomes were originally characterized through RNA sequencing of diverse insects and likely represent insect-specific viruses (*Kobayashi et al., 2013*; *Remnant et al., 2017*; *Shi et al., 2016*; *Teixeira et al., 2016*). The (+) ssRNA viral poxin sequences are highly divergent, sharing between 10–25% identity with one another, and occur in variable genome positions including at the extreme N-terminus and up to 1000 amino acids inside the predicted viral polyprotein (*Figure 5A*, *Figure 5—figure supplement 1*). Alignment of the (+)ssRNA viral poxin sequences reveals conservation of putative protease cata- lytic residues (histidine, aspartate, and serine) (*Figure 5B*), with the catalytic serine residing in an

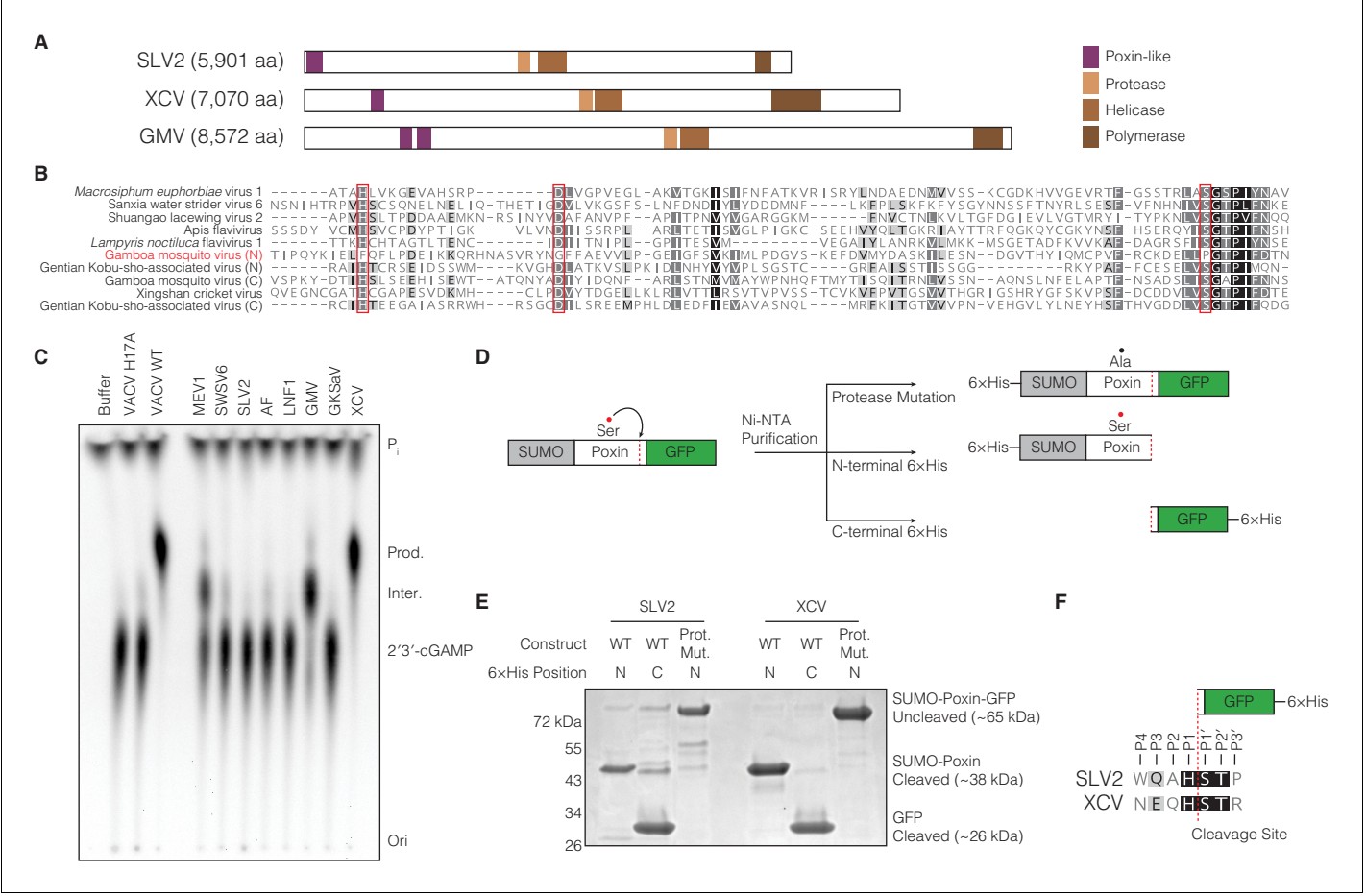

**Figure 5.** (+)ssRNA viruses encode functional self-cleaving poxin nucleases. (**A**) Diagrams of three representative amarillovirus polyproteins with poxin- like regions identified by PSI-BLAST highlighted in purple. Poxin is encoded in (+)ssRNA viral genomes at the extreme N-terminus or up to ~1000 amino acids inside the polyprotein, and is duplicated in some viruses like Gamboa mosquito virus (*Figure 5—figure supplement 1*). (**B**) Alignment confirming strong conservation of putative protease catalytic residues (red boxes) within amarillovirus poxins. Duplicated poxin proteins are denoted (N) or (C) indicating their positions in the polyprotein relative to one another. (**C**) TLC analysis of 2′3′-cGAMP degradation by recombinant amarillovirus poxin proteins (*Figure 5—figure supplement 2*). Amarillovirus poxin proteins from *Macrosiphum euphorbiae* virus 1, Gamboa mosquito virus, and XCV retain 2′3′-cGAMP nuclease activity. (**D**) Schematic showing constructs used to assess autoproteolytic cleavage activity of two amarillovirus poxin homologs from SLV2 and XCV. (**E**) SLV2 and XCV poxin proteins are active proteases. Self-cleavage during expression and purification allows separation of a SUMO-poxin fragment from a C-terminal GFP tag. Mutation to the amarillovirus poxin predicted protease catalytic site disrupts all detectable cleavage. (**F**) Edman degradation mapping identifies a conserved cleavage site in the C-terminus of SLV2 poxin and XCV poxin demonstrating strict recognition of an H/ST motif for autoproteolytic processing, despite these enzymes sharing only 21% identity. All data are representative of two independent experiments.

The online version of this article includes the following figure supplement(s) for figure 5:

**Figure supplement 1.** Amarillovirus genome structure and cloning of poxin-like region.

**Figure supplement 2.** Biochemical analysis of amarillovirus poxin protease and nuclease activity.

SGTP motif that is consistent with the vestigial AGTP motif conserved in the lepidopteran poxin protease active site (*Figure 5B*, *Figure 4—figure supplement 1D*).

We cloned putative poxins from the eight identified amarilloviruses and expressed them in *E. coli* as SUMO-fusion recombinant proteins (*Figure 5C*, *Figure 5—figure supplement 2A*). Activity analysis demonstrates that (+)ssRNA viral poxins from *Macrosiphum euphorbiae* virus 1, Gamboa mosquito virus, and Xingshan cricket virus (XCV) efficiently catalyze 2′3′-cGAMP degradation verifying these proteins as functional poxin family members. Given the high degree of divergence exhibited by amarillovirus poxins, we compared the specificity of XCV poxin for 2′3′-cGAMP and for a cGAMP isomer with only 3′–5′ linkages (3′3′-cGAMP). XCV poxin retains a high degree of specificity for 2′3′-cGAMP, similar to poxins tested from within each other group of enzymes (*Figure 5—figure supplement 2B*). To assess if amarillovirus poxins possess autoproteolytic cleavage activity, we focused on the XCV poxin enzyme capable of 2′3′-cGAMP cleavage and a divergent homolog from Shuangao lacewing virus 2 (SLV2) that readily expressed to high levels in *E. coli* (*Figure 5—figure supplement 2A*). The XCV and SLV2 poxins were each fused to a C-terminal GFP tag and purified with either an N-terminal or C-terminal 6 × His tag to assess self-cleavage (*Figure 5D*). Purification of the XCV or SLV2 poxins with an N-terminal tag yielded a fragment corresponding in size to a SUMO-poxin fusion, demonstrating proteolytic cleavage and removal of the GFP (*Figure 5E*). Likewise, purification with a C-terminal 6 × His tag confirmed these results and yielded a smaller fragment corresponding only to the cleaved GFP tag. Mutation of the putative protease catalytic serine residue conserved between amarillovirus poxin proteins blocked all cleavage (*Figure 5E*).

Using Edman degradation, we mapped the XCV and SLV2 cleavage motifs to an identical H/ST sequence conserved in both viruses (*Figure 5F*). Removal of the mapped cleavage site from the XCV poxin–GFP fusion construct abrogates all detectable proteolysis, confirming this motif is required for self-cleavage (*Figure 5—figure supplement 2C*). Further, mutation of individual residues in the XCV poxin cleavage site motif demonstrates that the histidine residue at position P1 directly N-terminal to the scissile bond is critical for cleavage site recognition (*Figure 5—figure supplement 2C*). A search of the XCV polyprotein for the cleavage site motif NxQH (*Figure 5F*) reveals that this motif occurs in only two instances across the entire 7070 amino-acid polyprotein, at the mapped C-terminal cleavage site and again just N-terminal to the poxin-like region (*Figure 5—figure supplement 2D,E*). Conservation of this motif at only these two positions in the polyprotein suggests that XCV poxin may be adapted for self-excision at both the N- and C-termini (*Figure 5—figure supplement 2F*). Notably, the ability of XCV poxin to catalyze both auto-proteolysis and nucleolytic cleavage of 2′3′-cGAMP (*Figure 5C,E*) confirms the existence of functional self-cleaving poxin enzymes within insect (+)ssRNA viruses. Together, these data verify a model for poxin evolution and demonstrate that this family of immune evasion proteins diverged from a self-cleaving accessory protease (*Figure 5—figure supplement 2F*).

## Discussion

Our results reveal that poxins are a widespread family of enzymes dedicated to 2′3′-cGAMP degradation and control of cGAS-STING immunity. Through biochemical and structural analysis, we reconstruct the evolutionary history of poxins and define a clear molecular connection with viral protease enzymes. Structures of lepidopteran poxins reveal unexpectedly strong homology with serine proteases from (+)ssRNA viruses and explain how poxins originated from a self-cleaving viral protease that gained a secondary nuclease active site for 2′3′-cGAMP cleavage (*Figure 3*). In this model, acquisition of a C-terminal domain enabled protease dimerization and creation of a new binding site for 2′3′-cGAMP recognition (*Figure 6*). Substrate contortion within this pocket catalyzes a metal-independent reaction that degrades 2′3′-cGAMP and potently inhibits host antiviral immunity (*Figure 2*).

A structure-guided maximum-likelihood tree of all poxin sequences identified in our study provides a global view of poxin diversity and the prominent role of antagonism of host cGAS-STING signaling in mammalian immunity and insect viral replication (*Figure 6A*, *Figure 6—figure supplement 1*). The currently available poxin enzyme sequences form nine groups or subgroups based on phylogenetic analysis and species origin. Of note, amarillovirus poxin sequences in Group five and baculovirus poxin sequences in Group 2C do not achieve >50% bootstrap support. As new sequences become available, future bioinformatic work will be required to reconstruct the exact mechanism of

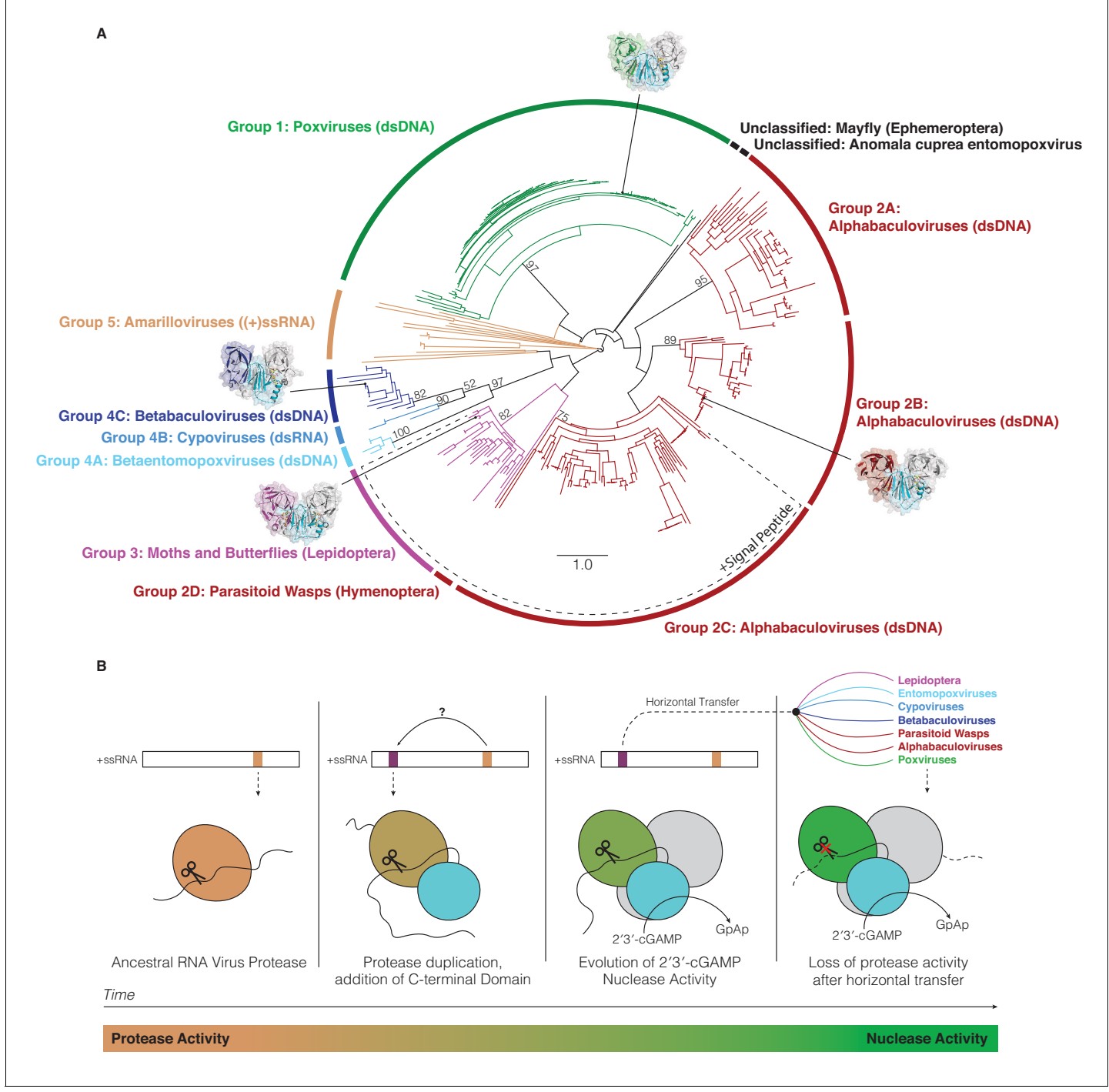

**Figure 6.** Horizontal transfer and evolution of poxin enzymes from proteases to nucleases. (**A**) Structure-guided phylogenetic tree depicting global poxin diversity. 100 bootstrap replicates were performed, and branch support is provided for all poxin groups with support values > 50. Most poxin sequences within groups highlighted with a dotted line encode putative signal peptides, suggesting extracellular or secretory roles of some alphabaculovirus and lepidopteran poxin homologs. Poxin enzymes with crystal structures are highlighted with black dots. Unclassified poxins from the mayfly *E. danica* and *Anomala cuprea* entomopoxvirus were grouped together on the tree for simplicity but are not closely related to one another. The tree is visualized here as rooted to emphasize and enumerate global poxin sequence diversity, and the unrooted visualization is available in *Figure 6—figure supplement 1*, annotated with all bootstrap values for deep branches. (**B**) Model providing a possible explanation for the evolution of poxin proteins from (+)ssRNA viral proteases. Transition from protease to nuclease activity occurred through evolution of a second active site, and horizontal transfer to other viruses and insect hosts.

The online version of this article includes the following figure supplement(s) for figure 6:

*Figure 6 continued on next page*

*Figure 6 continued*

**Figure supplement 1.** Unrooted visualization of structure-based maximum-likelihood poxin phylogeny.

poxin horizontal spread, and to confirm RNA viruses as the progenitors of all poxin enzymes. Our phylogenetic analysis further emphasizes that nearly all insect viruses which encode a poxin enzyme exclusively infect moths and butterflies in the order Lepidoptera. As lepidopteran hosts also encode endogenized versions of poxin, our results reveal moths and butterflies and their pathogens as an epicenter for extensive radiation of this protein family. The most divergent poxin enzyme sequences are encoded in the genomes of (+)ssRNA viruses from the order *Amarillovirales*, distantly related to mammalian (+)ssRNA pathogens like dengue virus and hepatitis C virus. Identification of functional self-cleaving poxin enzymes with 2′3′-cGAMP nuclease activity in circulating amarilloviruses provides further support for the biochemical model of evolution of poxin enzymes from (+)ssRNA viral pro-teases. Horizontal transfer from an ancient amarillovirus likely seeded poxin diversification in Lepi-doptera and eventual acquisition by mammalian poxviruses. Amarillovirus genomes encoding poxin have been identified in geographically distant locations and from extremely diverse insect species representing six different phylogenetic orders (*Faizah et al., 2020*; *Kobayashi et al., 2013*; *Kondo et al., 2020*; *Remnant et al., 2017*; *Shi et al., 2016*; *Teixeira et al., 2016*). As our under-standing of viral diversity continues to broaden, it is likely that additional immune evasion mecha-nisms will be identified as shared between mammalian viruses and invertebrate pathogens.

In mammals, poxin functions to degrade 2′3′-cGAMP and block induction of antiviral signaling by the cGAS-STING pathway during poxvirus infection (*Eaglesham et al., 2019*). In most mammalian poxviruses, poxin exists as a fusion to a C-terminal domain with homology to mammalian schlafens (*Eaglesham et al., 2019*). Recent work with ectromelia virus demonstrates the schlafen domain is dispensable for evasion of cGAS-STING signaling, and that mammalian schlafens fail to complement poxin–schlafen deletion mutant viruses (*Hernáez et al., 2020*). Given conservation of the poxin–schlafen fusion amongst most orthopoxviruses (*Eaglesham et al., 2019*; *Hernáez et al., 2020*), fur-ther work is required to explore auxiliary roles for the schlafen domain in cGAS-STING regulation by poxin.

The widespread distribution of poxin enzymes in insect viruses supports a prominent role for 2′3′-cGAMP-signaling in metazoan antiviral immunity. Recent studies of cGAS-STING signaling in insects have demonstrated that *Drosophila* STING drives NF-κB and autophagy signaling to restrict viral infection (*Goto et al., 2018*; *Gui et al., 2019*; *Hua et al., 2018*; *Liu et al., 2018b*; *Martin et al., 2018*). However, the upstream signaling machinery that activates STING in insects remains poorly understood. Insects encode enzymes like *Drosophila melanogaster* CG7194 and CG12970 that are part of the cGAS/DncV-like nucleotidyltransferase (CD-NTase) family (*Kranzusch, 2019*; *Whiteley et al., 2019*), but these enzymes are significantly divergent from mammalian cGAS and it is unclear if insect CD-NTases synthesize 2′3′-cGAMP or respond to cytosolic DNA. Our results show that nearly all poxin representatives from across all groups retain specificity for 2′3′-cGAMP and fail to cleave 3′3′-cGAMP. Although exceptions to poxin specificity exist (*Figure 5—figure supplement 2B*), these results suggest that 2′3′-cGAMP is a predominant ligand in insect immune signaling.

While most metazoan poxin enzymes were identified in the genomes of moths and butterflies, several examples suggest that poxins play an important role in immunity in diverse insects. Two dif-ferent parasitoid wasp species encode poxin homologs (*Glypta fumiferanae*: AKD28026 and *Micro-plitis demolitor*: XP_008552911), and previous work suggests these proteins may even play a role in parasitism of caterpillars, within which the wasps lay their eggs (*Béliveau et al., 2015*; *Burke et al., 2018*; *Strand and Burke, 2015*). Further, the mayfly *Ephemera danica* encodes a poxin homolog (KAF4524375), which possesses an intact HYK catalytic triad indicating functional poxin nuclease activity, but this enzyme fails to cluster within other poxin groups (*Figure 6A*). Future studies of these, and other insect poxin proteins will provide further insight into the ancient evolutionary rela-tionship between RNA-virus proteases and poxin nucleases.

Whereas insect viral poxins likely restrict immune activation similar to the function of poxin in mammalian poxviruses, the biological roles of poxin enzymes endogenized in the genomes of insects are less clear. One hypothesis is that insect poxins function to limit the magnitude of STING activa-tion. In agreement, our results suggest that host insect poxins function with kinetics distinct from

viral poxin counterparts (*Figure 2—figure supplement 2*). Although viral poxins are capable of degrading 2′3′-cGAMP even at low ligand concentrations, host poxins instead appear tailored for setting an upper threshold for the immune response. In mammals, a growing body of work suggests that 2′3′-cGAMP can be released from infected cells and imported to activate bystander immunity (*Carozza et al., 2020*; *Luteijn et al., 2019*; *Ritchie et al., 2019*; *Zhou et al., 2020*). Interestingly, our bioinformatic analysis demonstrates that many baculoviruses encode two different poxin enzymes with one variant containing a signal peptide for extracellular secretion (*Figure 6A*; *Craveiro et al., 2015*). Likewise, host lepidopteran poxins are encoded as multiple isoforms with and without a signal peptides (e.g. *Trichoplusia ni* XP_026730193 and XP_026730202) (*Supplementary file 1*; *Chen et al., 2019*). In insects, poxins may therefore regulate extracellular 2′3′-cGAMP signaling in addition to controlling cytosolic cGAS-STING activation.

In contrast to the widespread distribution of poxin enzymes among insects and insect viruses, there is a puzzling lack of cytosolic 2′3′-cGAMP nuclease machinery in mammalian cells (*Eaglesham et al., 2019*). The only enzyme known to degrade 2′3′-cGAMP in humans is the nuclease ENPP1, which is exclusively extracellular and regulates signaling outside of the cell (*Carozza et al., 2020*; *Li et al., 2014*). Functional homologs of ENPP1 have not been identified in insects, indicating that alternative mechanisms for regulating cGAS-STING signaling may exist in these animals. Our data indicate that insect poxins may be expressed as both secreted and cytosolic forms, perhaps having functional roles in both intra- and extracellular cGAS-STING regulation. This abundance of enzymes that efficiently degrade 2′3′-cGAMP in the cytosol of insects may have provided the opportunity for poxviruses to acquire a new mechanism of immune control that did not exist in mammalian cells. Poxins have traversed a vast evolutionary distance from an origin in insect (+)ssRNA viruses to a role in enabling mammalian poxvirus pathogens to evade cGAS-STING immunity, and a remarkable transition from protease to nuclease activity provides a clear example of how proteins can evolve through gain and loss of enzymatic function. Acquisition of poxins from insect viruses further underscores the functional similarities between mammalian and insect innate immunity and reveals the importance of environmental genetic diversity as a driver for evolution of pathogenic viruses.

## Materials and methods

### Bioinformatic identification and cloning of poxin homologs

VACV poxin and the host lepidopteran *T. ni* poxin sequences were used to initiate queries of the NCBI nonredundant protein database using position-specific iterative BLAST (PSI-BLAST) (*Altschul et al., 1997*) on January 17, 2020. Additional searches were performed on May 27, 2020 using the mapped boundaries for SLV2 (M1–H227) and XCV (C782–H1007) poxins, along with VACV and *T. ni* poxin as queries, allowing identification of 18 additional poxin sequences. For each analysis, continued iterations between 6 and 10 rounds were run until convergence of results. The BLAST default settings were used, specifying a PSI-BLAST E-value cutoff of 0.005 for inclusion in the next search round, with BLOSUM62 scoring matrix, and gap costs set at existence: 11, extension: 1. Initial results for VACV and *T. ni* query proteins were combined for a total of 351 poxin homolog sequences, and were largely overlapping with two sequences identified only with VACV as a query, and 26 identified only using *T. ni* poxin as a query. Results obtained in our second analysis using SLV2 and XCV poxin sequences as additional queries included 18 sequences not previously identified, for a total of 369 poxin homologs (*Supplementary file 1*). Some proteins identified here as poxin enzymes have previously been referred to by other names, such as p26 in alphabaculoviruses, HDD13 in Lepidoptera, Schlafen in poxviruses (in most orthopoxviruses, poxin sequences are fused to a C-terminal schlafen domain, but in some cases this name has carried over to poxin proteins in other poxviruses which are unfused and have no homology to mammalian schlafens), and acetyltransferase-like protein in betabaculoviruses. Proteins smaller than 179 amino acids or larger than 532 amino acids, such as sequences identified within large RNA-virus polyproteins, appeared to represent sequence fragments, or proteins too large to be a poxin protein alone and were excluded from our initial poxin biochemical screen. Of the remaining sequences, 33 representative enzymes were selected for biochemical analysis. To study amarillovirus poxin proteins, soluble fragments from within the viral polyprotein were identified using an estimated boundary of ~200 amino acids around regions of poxin homology, individual analysis of protein disorder prediction with DisoPred3

(*Ward et al., 2004*), and homology modeling to lepidopteran poxin structures with Phyre2 (*Kelley et al., 2015*). Refseq accession numbers for the poxin proteins in biochemical screen in *Figure 1B* are in order as follows: vaccinia virus, VACV (YP_233066.1); monkeypox virus, MPXV (AAY97169.1); cowpox virus, CPXV (NP_619978.1); akhmeta virus, AKHV (AXN74977.1); volepox virus, VPXV (YP_009281928.1); raccoonpox virus, RPXV (YP_009143488.1); *M. sanguinipes* entomopoxvirus, MSEV (NP_048308.1); sea otter poxvirus, SOPV (YP_009480542.1); pteropox virus, PTPV (YP_009480542.1); *Anomala cuprea* entomopoxvirus, ACEV (YP_009001652.1); *Glypta fumiferanae* (YP_009001652.1); *Microplitis demolitor* (XP_008552911.1), *Lymantria dispar* nucleopolyhedrovirus, LdNPV (AIX47878.1), *Chrysodeixis includens* nucleopolyhedrovirus, CiNPV-A (AOL57023.1); *Spodoptera frugiperda* nucleopolyhedrovirus, SfNPV-A (YP_001036423.1); *Choristoneura rosaceana* nucleopolyhedrovirus, CrNPV-A (YP_008378498.1); *Bombyx mori* nucleopolyhedrovirus, BmNPV (NP_047534.1); *Autographa californica* nucleopolyhedrovirus, AcNPV (NP_054166.1); *Choristoneura rosaceana* nucleopolyhedrovirus, CrNPV-B (YP_008378375.1); *Spodoptera frugiperda*, SfNPV-B (YP_001036378.1); *Trichoplusia ni* nucleopolyhedrovirus, TnNPV-B (YP_308949.1); *Lambdina fiscellaria* nucleopolyhedrovirus, LfNPV (YP_009133287.1); *Ectropis obliqua* nucleopolyhedrovirus, EoNPV-B (YP_874245.1); *Pieris rapae* (XP_022120734.1); *Bombyx mori* (XP_021205460.1); *Papillio xuthus* (KPI98967.1); *Helicoverpa armigera* (XP_021193734.1); *Trichoplusia ni* (XP_026730193.1); *Danaus plexippus* (OWR54847.1); *Dendrolimus punctatus* cypovirus 22, DpCV 22 (YP_009111323.1); *Inachis io* cypovirus 2, IiCV 2 (YP_009002595.1); *Pieris rapae* Granulovirus, PrGV (AHD24842.1); *Amsacta moorei* entomopoxvirus, AMEV (NP_065007.1). Poxin-like regions of amarillovirus polyproteins were cloned according to boundaries defined in *Figure 5—figure supplement 1* from the following accessions: *Macrosiphum euphorbiae* virus 1, MEV1 (NC_028137.1); Sanxia water strider virus 6, SWSV6 (NC_028368.1); Shuangao lacewing virus 2, SLV2 (NC_028373.1); *Apis* flavivirus, AF (NC_035071.1); *Lampyris noctiluca* flavivirus 1, LNF1 (MH620810.1); Gamboa mosquito virus, GMV (NC_028374.1); Gentian Kobu-sho-associated virus, GKSaV (NC_020252.1); Xingshan cricket virus, XCV (NC_028370.1). Sequences encoding each protein were not codon optimized for *E. coli* expression unless necessary for gene synthesis. Genes encoding the following proteins were codon optimized: *T. ni* poxin, AMEV poxin, and LdNPV poxin. Predicted signal peptides were removed before gene synthesis according to the cleavage site prediction with SignalP 5.0 (*Almagro Armenteros et al., 2019*).

## Protein alignments and phylogenetic trees

All protein alignment diagrams were created using the MAFFT FFT-NS-i iterative refinement method (*Katoh and Standley, 2013*), rendered in Geneious Prime (2020.0.5) and exported for annotation in Adobe Illustrator 24.1. Phylogenetic trees were constructed in Geneious Prime, using alignments made using MAFFT FFT-NS-i iterative refinement (*Figures 1* and *2*) or PROMALS3D (*Pei et al., 2008*; *Figure 6*, *Figure 6—figure supplement 1*). The tree in *Figure 1* was produced from a sequence alignment of 33 poxin proteins selected for initial biochemical analysis using the neighbor-joining method and Jukes-Cantor genetic distance model with no outgroup and rendered using proportionally transformed branches for alignment to TLC images in Illustrator. To produce the phylogeny of all poxin enzymes in *Figure 6*, we aligned all poxin sequences ranging in size from 179 to 397 with poxin monomer crystal structures for VACV poxin (6EA9), AcNPV poxin (6XB3), PrGV poxin (6XB4) and *T. ni* poxin (6XB5) using PROMALS3D (*Pei et al., 2008*). In order to place amarillovirus poxin-like sequences on the tree, sequence boundaries were predicted using the mapped cleavage sites of the SLV2 and XCV poxin-like regions. For both, these were 112 amino acids N-terminal and 115–116 amino acids C-terminal to the serine in the conserved SGxP motif. Therefore, these boundaries were applied to each amarillovirus poxin sequence to adjust for the wide variety of sequence lengths identified by PSI-BLAST. For Lepidoptera, a single isoform of each poxin protein was included in the alignment used to generate the tree. In poxviruses, many representatives of poxin are encoded as C-terminal fusions to a domain with homology to mammalian schlafen proteins. For these sequences, only the region corresponding to the poxin domain was included in the alignment, and the schlafen domain was removed, extending from the conserved amino-acid motif LLNSGGG to the C-terminus. The resulting PROMALS3D alignment was subjected to maximum-likelihood analysis with 100 bootstrap replicates using PhyML in order to generate the final poxin phylogeny (*Guindon et al., 2010*; *Pei et al., 2008*).

## Protein expression and recombinant protein purification

All poxin protein constructs were cloned into a custom pET vector designed to express an N-terminal 6 × His tagged SUMO2 fusion (*Zhou et al., 2018*) in *E. coli*, using synthetic DNA fragments (IDT) and NEBuilder HiFi DNA Assembly mix (NEB). Certain constructs produced for the amarillovirus poxin self-cleavage experiment in *Figure 5E* were cloned into an alternative custom pET vector with an N-terminal SUMO2 fusion and C-terminal 6 × His tag (*Whiteley et al., 2019*). Recombinant proteins were produced in the *E. coli* BL21 RIL strain (Agilent), in 2 ml (small scale) or 50 ml (large scale) MDG starter cultures, before growth and induction in 10 ml (small scale, for biochemistry) or 1 L (large scale, for kinetics assays and crystallography) M9ZB cultures as previously described (*Zhou et al., 2018*). Selenomethionine (SeMet)-labeled proteins for crystallography were grown in modified M9ZB medium as previously described (*Eaglesham et al., 2019*). Cells were collected by centrifugation, disrupted by sonication, and recombinant protein was purified using Ni-NTA beads (Qiagen) as previously described (*Eaglesham et al., 2019*). Poxin proteins purified for biochemical assays in *Figure 1B*, *Figure 5C,E*, *Figure 1—figure supplement 1B*, *Figure 2—figure supplement 1B*, and *Figure 5—figure supplement 2B* were buffer-exchanged after elution from Ni-NTA beads, and stored in 20 mM HEPES-KOH pH 7.5, 250 mM KCl, 10% glycerol, 1 mM TCEP-KOH without removal of the SUMO2 tag. Proteins used for crystallography and enzyme kinetics were dialyzed overnight into a buffer composed of 20 mM HEPES-KOH pH 7.5, 250 mM KCl, and 1 mM TCEP-KOH with approximately 250 ng hSENP2 protease (fragment with D364–L589 with M497A mutation) (*Reverter and Lima, 2006*). Untagged poxin proteins were then further purified using 16/600 S75 or S200 size-exclusion chromatography columns (GE) in the same buffer, concentrated for storage and cryoprotection, flash frozen in liquid nitrogen, and stored long-term at −80°C. Purified proteins were resolved on 4–20% Mini-Protean TGX gels (Bio-Rad) according to manufacturers' specifications and stained with Coomasie G-250 (VWR).

## Synthesis of cyclic dinucleotides

2′3′-cGAMP for poxin nuclease assays was synthesized using the mouse cGAS catalytic domain P147–L507. Recombinant human SUMO2-tagged mouse cGAS was expressed in *E. coli* and purified using Ni-NTA affinity chromatography as previously described (*Zhou et al., 2018*). Briefly, the SUMO2 tag was removed as described above and mcGAS was further purified with heparin ion-exchange and S75 size-exclusion chromatography. Mouse cGAS (5 μM) was incubated for 2 hr at 37°C with 200 μM ATP and 200 μM GTP in the presence of 2 μM 45 bp stimulatory dsDNA in a 20 μl reaction with final buffer composed of 50 mM HEPES-KOH pH 7.5, 5 mM Mg(OAc)$_2$, 37.5 mM KCl, and 1 mM DTT. Reactions were trace-labeled with [α-$^{32}$P] GTP (Perkin-Elmer). Reactions were terminated through addition of 1 μl Quick CIP (NEB) to digest remaining nucleoside triphosphate substrates, heat inactivated for 5 min at 80°C, and frozen at −20°C before use in nuclease assays. 3′3′-cGAMP was enzymatically synthesized in a similar manner using recombinant *Vibrio cholerae* DncV incubated with ATP and GTP as previously described (*Eaglesham et al., 2019*; *Kranzusch et al., 2014*). All 3′–5′ linked cyclic dinucleotides were prepared with 200 μM ATP and 200 μM GTP.

2′3′-cGAMP used for crystallography was enzymatically synthesized and purified as previously described (*Eaglesham et al., 2019*), by incubation of 100 nM recombinant mouse cGAS with 500 μM each ATP and GTP substrates and 50 μg ml$^{-1}$ salmon sperm DNA in reaction buffer (10 mM Tris-HCl pH 7.5, 12.5 mM NaCl, 10 mM MgCl$_2$, 1 mM DTT) at 37°C for 24 hr. 2′3′-cGAMP was then purified by ion-exchange (2 × 5 ml HiTrap Q columns) using a gradient of 0–2 M NH$_4$OAc. Eluted 2′3′-cGAMP was freeze-dried and washed twice with methanol before final lyophilization and storage as powder at −20°C.

## Poxin nuclease activity assays

Poxin nuclease assays were performed as previously described (*Eaglesham et al., 2019*). Reactions were carried out at 37°C in 10 μl buffer (50 mM HEPES-KOH pH 7.5, 40 mM KCl, 1 mM DTT) with 1 μl of a cGAS 2′3′-cGAMP synthesis reaction (~20 μM final concentration of 2′3′-cGAMP). Reactions for the nuclease activity screens in *Figure 1B* and *Figure 5C* were carried out using 1 μl of buffer-exchanged Ni-NTA elutions for each recombinant protein without normalization for protein concentration. Reactions with poxin active-site mutants in *Figure 2—figure supplement 1B* and reactions testing specificity of diverse poxins in *Figure 5—figure supplement 2B* were carried out using 1 μl

of a 1 μM stock for each recombinant protein, and incubated for 15 min. For the poxin nuclease activity screen in *Figure 1B*, reactions were incubated for 1 hr, and reactions with amarillovirus poxin proteins in *Figure 5C* were performed for 20 hr. Longer reactions were used to allow more sensitive detection of 2′3′-cGAMP degradation activity. All reactions were terminated by spotting on a PEI cellulose thin-layer chromatography plate (EMD Millipore), and reaction products were resolved using a TLC mobile phase composed of 1.5 M $KH_2PO_4$ pH 3.8. After developing, TLC plates were dried and exposed to a phosphor screen overnight before imaging on a typhoon phosphor-imager (GE). TLC images were cropped and adjusted for brightness and contrast in Fiji (Version 2.0.0-rc-69–1.52 p).

## Poxin Michaelis-Menten kinetic analysis

In order to study poxin enzyme kinetics, poxin nuclease activity assays were carried out using stocks of chemically-synthesized 2′3′-cGAMP (Biolog) mixed with a small amount of [$^{32}$P]−2′3′-cGAMP tracer to achieve defined substrate concentrations. [$^{32}$P]−2′3′-cGAMP tracer was produced in 10 μl reactions with mouse cGAS as detailed above, using 3 μl [α-$^{32}$P] GTP (Perkin-Elmer,~10 μM final concentration) and 10 μM ATP. After a 2 hr incubation at 37°C, quenching with Quick CIP, and heat inactivation for 5 min at 80°C, the tracer was diluted 1:5 (50 μl final volume) in RNase-free water (VWR). Chemically-synthesized 2′3′-cGAMP was then re-suspended in water, and labeled by addition of [$^{32}$P]−2′3′-cGAMP tracer at a 1:50 dilution to achieve radioactively labeled stocks at a range of concentrations. 10 μl reactions were assembled in triplicate in 8-well strips with one well serving as a tracer-only background control, and seven wells serving as experimental poxin degradation reactions with varying concentrations of substrate. Reactions were pre-warmed to 37°C in a 96-well heat block for 5 min prior to addition of 1 μl of buffer to the tracer-only background control and 1 μl of poxin protein to the experimental wells. Reactions were mixed with a multi-channel pipettor, and stopped by spotting directly onto a TLC plate after 30 s. VACV poxin reactions were carried out with 20 nM protein (10 nM enzyme dimer) incubated with 2′3′-cGAMP at the following final concentrations: 0.1, 0.25, 0.5, 0.75, 1, 2.5, and 5 μM. AcNPV poxin reactions were carried out with 20 nM protein (10 nM enzyme dimer) incubated with 2′3′-cGAMP at the following final concentrations: 0.25, 0.5, 0.75, 1, 2.5, 5, and 10 μM. *T. ni* poxin reactions were carried out with 100 nM protein (50 nM enzyme dimer) incubated with 2′3′-cGAMP at the following final concentrations: 50, 100, 150, 250, 500, 750, and 1 mM. Reaction progress was monitored using thin-layer chromatography, and quantified using ImageQuant software (GE). Percentage 2′3′-cGAMP turnover was calculated by quantification of the cleaved 2′3′-cGAMP spot intensity divided by the total signal for cleaved and uncleaved 2′3′-cGAMP in each lane. This value was then adjusted by subtraction of the percentage turnover value observed for the tracer-only negative control. To obtain the initial rates of 2′3′-cGAMP degradation in μM min$^{-1}$, adjusted percent turnover was multiplied by the total concentration of 2′3′-cGAMP in each reaction, and divided by the length of the reaction (0.5 min). 2′3′-cGAMP dependent enzyme kinetics were fit using the Michaelis-Menten model in GraphPad Prism, and $K_{cat}$ values were determined using the concentrations of poxin dimer, the minimal active enzyme unit. Results for each enzyme shown in *Figure 2—figure supplement 2* are a single experiment (n = 3 technical replicates), representative of at least two biological replicates.

## Electrophoretic mobility shift assay

Stable poxin–2′3′-cGAMP complex formation was assessed using an electrophoretic mobility shift assay as previously developed for the receptor STING (*Morehouse et al., 2020*; *Whiteley et al., 2019*). All 2′3′-cGAMP-binding experiments were performed with catalytic inactive poxin mutants VACV poxin H17A, AcNPV poxin H46A, and *T. ni* poxin H56A to prevent cleavage. Briefly, radiolabeled 2′3′-cGAMP was diluted to a final concentration of ~50 nM into 10 μl reactions containing 1 × reaction buffer (50 mM KCl, 50 mM Tris-HCl pH 7.5, and 1 mM TCEP) and 0–20 μM recombinant poxin protein as indicated. Reactions were incubated for 30 min at 25°C, then separated on a 7.2 cm 6% nondenaturing polyacrylamide gel run at 100 V for 45 min in 0.5 × TBE buffer. The gel was fixed for 15 min in a solution of 40% ethanol and 10% acetic acid before drying at 80°C for 1 hr and then exposed to a phosphor screen and imaged with a Typhoon Trio Variable Mode Imager (GE Healthcare). Signal intensity was quantified using ImageQuant 5.2 software (GE Healthcare) and analyzed in GraphPad Prism 8.4.3 using the specific binding with hill slope model to determine $K_D$. Note that

use of proteins with catalytically inactivating mutations likely results in an underestimated $K_D$ value for each poxin studied.

## Crystallization and structure determination

Crystals of AcNPV, PrGV, *T. ni* and *D. plexippus* poxin proteins were grown using hanging-drop vapor diffusion at 18˚C. AcNPV, PrGV, and *T. ni* poxin proteins were crystallized at a concentration of 7 mg ml$^{-1}$ in the presence of 2.5–5 mM 2′3′-cGAMP, yielding post-reactive structures bound to a cleaved Gp[2′–5′]Ap[3′] molecule. *D. plexippus* poxin protein crystals were grown at 7 mg ml$^{-1}$ in the presence of 300 µM phosphorothioate nonhydrolyzable 2′3′-cGAMP Isomer 2 (Biolog), but failed to crystallize in complex with this nucleotide, instead yielding an apo structure. SeMet-labeled and native AcNPV poxin crystals grew in 100 mM sodium acetate pH 4.8–5.2, 35–39% PEG-400 and were cryoprotected in mother liquor. SeMet-substituted and native PrGV poxin crystals grew in 200 mM MgCl$_2$, 100 mM Tris-Cl pH 8.3–8.7, 21–25% PEG-3350, and were cryoprotected with mother liquor supplemented with 15% ethylene glycol. SeMet-substituted *D. plexippus* poxin crystals were grown in 200–250 mM calcium acetate, 17–21% PEG-3350, and cryoprotected in NVH oil. *D. plexippus* native crystals grew in 200 mM ammonium citrate, 25% PEG-3350. Native *T. ni* poxin crystals were grown in 100 mM HEPES-KOH pH 8.0, 32% Jeffamine ED-2001, and frozen without cryoprotection. X-ray diffraction data were collected at the Advanced Photon Source beamlines 24-ID-C and 24-ID-E.

X-ray crystallography data were processed with XDS and AIMLESS (*Kabsch, 2010*), using the SSRL autoxds script (A. Gonzales, Stanford SSRL). Experimental phase information for AcNPV, PrGV, and *D. plexippus* poxin proteins was determined using data collected from SeMet-substituted crystals. For SeMet-labeled AcNPV poxin, PrGV poxin, and *D. plexippus* poxin, heavy sites were identified using HySS in Phaser (*Adams et al., 2010*), initial maps were produced using SOLVE/RESOLVE (*Terwilliger, 1999*), followed by model-building in Coot (*Emsley and Cowtan, 2004*). For AcNPV poxin, a phase solution could only be found using data processed into the space group I4$_1$, with 20 sites identified using HySS. Following initial manual building in Coot, a partial unrefined model was used as a molecular replacement search which obtained a solution in the spacegroup P1 with 16 AcNPV poxin copies in the asymmetric unit arranged as a double-helical filament. Analysis of data pathologies after processing into the space group P1 using Xtriage within PHENIX showed a multivariate Z-score of 5.243, indicating twinning, and successful refinement was carried out in PHENIX with the twin operator -l,-h,h+k+l. Using SeMet-labeled PrGV poxin crystals, eight sites were identified using HySS, and an initial map was calculated as above followed by model-building and refinement in PHENIX. For *D. plexippus* poxin, HySS detected 10 sites, allowing calculation of an initial map, model-building in coot, and refinement in PHENIX. The *D. plexippus* poxin structure was subsequently used as a molecular replacement search model to determine an initial map of the related *T. ni* poxin (50% identical), followed by model-building in coot and refinement in PHENIX. All structure figures were rendered using PyMOL (version 2.3.3).

## Dali structural homology analysis

In order to compare poxin structural homology to proteins in the Protein Data Bank, poxin monomer structures were uploaded and used to query the DALI server (*Holm, 2019*). Z-scores for homologs less than 90% identical to one another (PDB90) were then plotted using GraphPad prism to compare the distribution and overall level of homology detectable between each poxin structure and proteins in the Protein Data Bank (*Figure 3A,B*). Hits identified for each poxin protein in the PDB were sorted into 'protease' or 'other' groups using a PDB advanced search. The search was constructed by searching for overlaps between PDB IDs identified with DALI for each poxin with those identified using the following terms: Enzyme classification names equaling 'Serine endopeptidases' or 'Cysteine endopeptidases' or 'trypsin' or 'chymotrypsin' or 'enteropeptidase', OR Annotation name – CATH equaling 'trypsin-like serine proteases', OR Structure title containing phrases 'protease', or 'Peptidase', OR Macromolecule name containing phrases 'protease', or 'peptidase', OR annotation identifier – CATH equaling '2.40.10.120'. In order to compare the level of homology between *T. ni* poxin and eukaryotic or viral proteases by phylogenetic group, a list of the homolog PDB codes returned by DALI after query with *T. ni* poxin were used to perform a Protein Data Bank advanced search, and filtered by phylogeny as stated in the figure (*Figure 3B*). PDB codes assigned to these

phylogenetic groups were then plotted using GraphPad prism to compare the global level of homology between *T. ni* poxin and proteases from different groups (*Figure 3B*). VACV poxin was the top hit in all DALI searches with new poxin structures with a Z-score of 18.1 for AcNPV, 15.7 for PrGV, and 16.9 for *T. ni*, but was excluded to allow analysis for distant protease homology.

### Edman degradation

C-terminally 6 × His tagged GFP fragments resulting from amarillovirus poxin self-cleavage were subjected to Edman degradation for cleavage site identification. GFP fragments were produced at large-scale and purified using Ni-NTA and S75 size-exclusion chromatography as described above. The cleaved GFP fragments were then resolved on a 15% SDS-PAGE gel, transferred to a PVDF membrane (Bio-Rad), and stained with Coomasie G-250 (VWR). Bands corresponding to the cleaved proteins were excised from the membrane and submitted for five cycles of Edman Degradation at the Tufts University Core Facility. The resulting assignments were made by the facility: SLV2 (S, T, P, R, R), XCV (S,T, No Call, S, K), both of which corresponded exactly to only one site within the SLV2 and XCV GFP fusion constructs used, marked in *Figure 5F*.

## Acknowledgements

The authors acknowledge R Eaglesham, A Lee, K Chat, and members of the Kranzusch lab for helpful comments and discussion. The authors thank W Zhou and B Lowey for assistance with purification of cGAS and 2′3′-cGAMP. The work was funded by the Richard and Susan Smith Family Foundation, a Cancer Research Institute CLIP Grant, the Pew Biomedical Scholars program, The Mark Foundation for Cancer Research, the Parker Institute for Cancer Immunotherapy (P.J.K.), and support through an NIH T32 Training Grant AI007245 (J.B.E.). X-ray data were collected at the Northeastern Collaborative Access Team (NE-CAT) beamlines 24-ID-C and 24-ID-E, and at the Berkeley Center for Structural Biology (BCSB) beamline 8.2.2 at the Advanced Light Source. NE-CAT beamlines 24-ID-C and 24-ID-E are funded by the NIGMS (P30 GM124165, P41 GM103403) and an NIH-ORIP HEI grant (S10 RR029205) and used resources of the DOE Argonne National Laboratory Advanced Photon Source (under Contract No. DE-AC02-06CH11357).

## Additional information

### Funding

| Funder | Grant reference number | Author |
|---|---|---|
| Richard and Susan Smith Family Foundation | | Philip J Kranzusch |
| Cancer Research Institute | Clinic and Laboratory Integration Program | Philip J Kranzusch |
| Pew Charitable Trusts | Biomedical Scholars Program | Philip J Kranzusch |
| The Mark Foundation for Cancer Research | Emerging Leader Award | Philip J Kranzusch |
| The Parker Institute for Cancer Immunotherapy | | Philip J Kranzusch |
| National Institutes of Health | T32 Training Grant AI007245 | James B Eaglesham |

The funders had no role in study design, data collection and interpretation, or the decision to submit the work for publication.

### Author contributions

James B Eaglesham, Conceptualization, Data curation, Formal analysis, Investigation, Visualization, Writing - original draft, Project administration; Kacie L McCarty, Data curation, Formal analysis, Validation, Investigation, Writing - review and editing; Philip J Kranzusch, Conceptualization, Supervision, Funding acquisition, Writing - original draft

## Author ORCIDs
James B Eaglesham (iD) https://orcid.org/0000-0002-5725-0743
Kacie L McCarty (iD) https://orcid.org/0000-0002-5174-1904
Philip J Kranzusch (iD) https://orcid.org/0000-0002-4943-733X

## Decision letter and Author response
Decision letter https://doi.org/10.7554/eLife.59753.sa1
Author response https://doi.org/10.7554/eLife.59753.sa2

# Additional files

## Supplementary files
• Supplementary file 1. Poxin sequences identified and protein constructs utilized in this study. All poxin sequences identified by PSI-BLAST are listed here, along with details for all sequences used to construct the phylogenetic trees in *Figure 1* and *Figure 6*, as well as the protein constructs produced for poxin biochemical activity screens in *Figure 1* and *Figure 5*.

• Supplementary file 2. Crystallographic statistics, related to *Figures 2–4* and 6. This file summarizes data collection, phasing, and refinement statistics for X-ray crystallography data. All crystallography datasets were collected from individual crystals. Values in parentheses are for the highest resolution shell.

• Transparent reporting form

## Data availability
Diffraction data have been deposited in the PDB under the accession codes 6XB3, 6XB4, 6XB5, and 6XB6.

The following datasets were generated:

| Author(s) | Year | Dataset title | Dataset URL | Database and Identifier |
|---|---|---|---|---|
| Eaglesham JB, McCarty KL, Kranzusch PJ | 2020 | Structure of AcNPV poxin in post-reactive state with Gp[2'-5']Ap[3'] | https://www.rcsb.org/structure/6XB3 | RCSB Protein Data Bank, 6XB3 |
| Eaglesham JB, McCarty KL, Kranzusch PJ | 2020 | Structure of PrGV poxin in post-reactive state with Gp[2'-5']Ap[3'] | https://www.rcsb.org/structure/6XB4 | RCSB Protein Data Bank, 6XB4 |
| Eaglesham JB, McCarty KL, Kranzusch PJ | 2020 | Structure of Trichoplusia ni poxin in post-reactive state with Gp[2'-5']Ap[3'] | https://www.rcsb.org/structure/6XB5 | RCSB Protein Data Bank, 6XB5 |
| Eaglesham JB, McCarty KL, Kranzusch PJ | 2020 | Structure of Danaus plexippus poxin cGAMP nuclease | https://www.rcsb.org/structure/6XB6 | RCSB Protein Data Bank, 6XB6 |

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
