## [Decision Letter]

**Acceptance summary:**

The work represents an advance in our understanding of an interesting class of enzymes found in both viruses and hosts involved in inhibition and/or regulation of the cGAS/STING intrinsic immune pathway. The manuscript reports an impressive combination of phylogenetics, biochemistry, and structural biology, which led to an intriguing model for the origins of poxins and poxin-like enzymes.

**Decision letter after peer review:**

Thank you for submitting your article "Structures of diverse poxin cGAMP nucleases reveal a widespread role for cGAS-STING evasion in host-pathogen conflict" for consideration by *eLife*. Your article has been reviewed by three peer reviewers, including Nels C Elde as the Reviewing Editor and Reviewer #1, and the evaluation has been overseen by Karla Kirkegaard as the Senior Editor.

The reviewers have discussed the reviews with one another and the Reviewing Editor has drafted this decision to help you prepare a revised submission.

All three reviewers agreed that the work represents an advance in our understanding of an interesting class of enzymes found in both viruses and hosts involved in inhibition and/or regulation of the cGAS/STING intrinsic immune pathway. An impressive combination of phylogenetics, biochemistry, and structural biology led to an intriguing model for the origins of poxins and poxin-like enzymes, which will likely be of broad interest.

Two main areas of shared concern emerged in the review that need to be addressed for the manuscript to be published.

First, additional phylogenetic work should be conducted using maximum-likelihood and/or bayesian approaches, which also includes bootstrapping analysis to give a better sense of how robust the relationships are between sequences, as noted in some form by all three reviewers and are reflected in the major points attached to the letter.

Second, the authors should consider reviewer comments related to the inhibition/regulation models of poxin and poxin-like activity in viruses and hosts. While new experiments in this area are not necessary for consideration for publication, appropriate changes to the text will help to clarify and balance the presentation.

Reviewer 1:

Eaglesham and colleagues build on their recent discovery of poxins, poxvirus encoded enzymes that degrade 2'3'-cGAMP a crucial messenger in cGAS/STING mediated intrinsic immunity. They use homology searches, phylogenetic analysis, biochemistry, and structural biology to build a case for the evolutionary history for poxin emergence in viruses of insects, insects, and poxviruses. They propose 4 classes of poxins and present an intriguing model for the history of poxin origins from self-cleaving proteases to become nucleases of 2'3'-cGAMP.

The combination of approaches nicely advances the earlier discovery of poxins and in addition to providing important new details about both inhibition (viruses) and regulation (insects) of cGAS through nuclease activity, the work also shows how combined biochem/evolution analysis can reveal otherwise cryptic relationships between highly diverged enzymes. Some of the phylogenetic analysis might be improved to give a better sense of the robustness of relationships between poxin variants given the massive divergence among the proteins.

1) The phylogenetic analysis appears to exclusively use neighbor-joining methodology and lacks bootstrap analysis, two potential shortcomings in gaining a robust view of the evolutionary relationships among poxin and poxin-related sequences. The analysis would be greatly improved by at least one other approach (for example a maximum-likelihood method like the one implemented in PhyML) as well as bootstrap analysis with values reported at least at the major nodes proposed to classify and sub-classify into 4 groups. Based on Figure 1 how did the authors decide to propose 4 clades versus for example 3 or 5?

2) A model for a single origin of poxins from self-cleaving proteases is presented. However, in Figure 6, the amarillovirus poxins, although all labeled in brown, appear in two groups, with one of them more closely related to poxvirus poxins. If the topology is correct (see point 1) this supports a model where poxviruses independently acquired the ancestral poxin compared to the insects/insect viruses. A closer examination of the phylogenetic relationships and the differences in amarillovirus poxins is warranted. On a related note, do the authors have ideas on poxin differences in poxviruses, for example the addition of a schlafen domain.

Reviewer 2:

Eaglesham et al. use a biochemical and structural approach to describe the evolutionary history of a family of enzymes known as poxins, which function as 2'3'-cGAMP nucleases. The authors first identify diverse poxin homologs in virus and metazoan species and provide structural and functional evidence of shared 2'3'-cGAMP nuclease activity despite low sequence similarity across clades. Using structural homology searches, the authors identify even more distantly related shared structural elements between the N-terminus protease domain of insect poxins and proteases from +ssRNA viruses. To confirm their hypothesis that poxins are evolutionarily related to self-cleaving +ssRNA viral proteases, they compare the crystal structures of +ssRNA viral protease and lepidopteran poxin and identify vestigial self-cleaving protease active sites that are broadly conserved across Lepidoptera. They further show that members of +ssRNA flavivirus-like viruses from the *Amarillovirales* order encode active poxin nucleases that possess self-cleaving protease ability. Finally, the authors provide a possible evolutionary trajectory for poxins and suggest poxins arose from ancestral +ssRNA viral proteases, acquired secondary nuclease activity, and spread via horizontal gene transfer into diverse viral and insect genomes.

Overall, this is a very interesting and well written manuscript. The conclusions are well supported by their data and are impactful. Some areas of concern that would improve the manuscript are below:

1) Based on the data provided in Figure 2—figure supplement 2, the authors propose that viral enzymes have evolved to have higher binding affinity for cGAMP than host enzymes. However, they are concluding this based on kinetic measurements of only two viral enzymes and only one host enzyme. Could the authors provide kinetic data for a member of clade 4 (viral) and/or a Hymenoptera member of clade 2 (presumably "host") to further support his claim?

2) In Figure 6, the authors use a neighbor-joining algorithm to generate phylogenetic trees that they then use to infer deep ancestral relationships of these diverse enzymes. It would be better to use maximum likelihood or Bayesian algorithms to generate these trees, as these approaches are better at resolving deep branches in trees.

3) Also in Figure 6, the authors speculate that an ancestral duplication of the viral protease led to the dual-function protease/poxin domains found in amarilloviruses. However, many viral genomes have proteases that are unrelated to each other and are instead distinct folds, etc. Are the N-terminal protease/poxin domain and the other protease in these genomes obviously related to each other? If not, the model should be more agnostic in terms of the exact origin of the protease/poxin domain.

4) The authors state in the Discussion that their data suggest that 2'3'-cGAMP is a predominant ligand in insect immune signaling. Have they shown that these divergent poxins, especially those from amarilloviruses, can exclusively hydrolyze 2'3'-cGAMP?

---

## [Author Response]

[…] Two main areas of shared concern emerged in the review that need to be addressed for the manuscript to be published.First, additional phylogenetic work should be conducted using maximum-likelihood and/or bayesian approaches, which also includes bootstrapping analysis to give a better sense of how robust the relationships are between sequences, as noted in some form by all three reviewers and are reflected in the major points attached to the letter.

We have performed the new phylogenetic analysis as requested, using a maximum-likelihood approach with bootstrapping, and Figure 6 has been updated with the resulting tree. Additional data regarding this analysis are available in Figure 6—figure supplement 1. This analysis further confirms nearly all groups of poxin enzymes proposed in the manuscript but shows that deep relationships between poxin enzymes cannot be fully resolved using the set of sequences currently available. While the inner branches of the poxin phylogeny remain unresolved, our biochemical and structural analysis of poxin proteins provides strong support for our model that these proteins originated as self-cleaving proteases in RNA virus genomes. As additional poxin sequences become available, we anticipate that our study will enable future bioinformatic analyses to resolve the deepest relationships between poxins and provide a more detailed explanation for the phylogenetic distribution of this protein family across different viruses and animal species.

Second, the authors should consider reviewer comments related to the inhibition/regulation models of poxin and poxin-like activity in viruses and hosts. While new experiments in this area are not necessary for consideration for publication, appropriate changes to the text will help to clarify and balance the presentation.

Thank you for these comments and helpful suggestions. As outlined below, we have addressed these points with new results including cyclic dinucleotide cleavage experiments to assess specificity of diverse poxin enzymes, protease experiments to further define amarillovirus poxin protease activity, and electrophoretic mobility shift assay experiments to directly measure 2′3′-cGAMP binding for VACV, AcNPV, and *T. ni* poxin proteins. We have additionally clarified discussion of alternative cGAS-STING regulatory strategies as suggested.

Reviewer 1:[…] Some of the phylogenetic analysis might be improved to give a better sense of the robustness of relationships between poxin variants given the massive divergence among the proteins.1) The phylogenetic analysis appears to exclusively use neighbor-joining methodology and lacks bootstrap analysis, two potential shortcomings in gaining a robust view of the evolutionary relationships among poxin and poxin-related sequences. The analysis would be greatly improved by at least one other approach (for example a maximum-likelihood method like the one implemented in PhyML) as well as bootstrap analysis with values reported at least at the major nodes proposed to classify and sub-classify into 4 groups. Based on Figure 1 how did the authors decide to propose 4 clades versus for example 3 or 5?

Thank you for this suggestion. We have extended our phylogenetic analysis to include a maximum-likelihood tree of diverse poxin sequences with bootstrap analysis as requested. These results provide strong bootstrap support for almost all clades proposed in the original manuscript, including clade 4 which is composed of enzymes from multiple groups of insect viruses. However, because not every proposed clade is well-supported with bootstrap value >50%, we have made changes to our manuscript for clarity. We now use the word “group” in place of the word “clade” to classify poxin enzymes, in order to emphasize that we are unable to confirm that each group is monophyletic, and to indicate that groups have been defined using both phylogenetic information and based on their species-origin. While most groups do constitute monophyletic clades, sufficient support is not achieved to designate monophyly of certain groups (Group 6: Amarillovirus poxins, and Group 2C: Baculovirus poxins). The amarillovirus poxins in Group 6 are of particular interest in the context of our biochemical and structural data. However, the comparatively small number of sequences available for this group, combined with the great divergence among amarillovirus poxins and their poorly defined N- and C-terminal boundaries, limit robust phylogenetic analysis and demonstrate the importance of our experimental approach in resolving deep evolutionary relationships.

2) A model for a single origin of poxins from self-cleaving proteases is presented. However, in Figure 6, the amarillovirus poxins, although all labeled in brown, appear in two groups, with one of them more closely related to poxvirus poxins. If the topology is correct (see point 1) this supports a model where poxviruses independently acquired the ancestral poxin compared to the insects/insect viruses. A closer examination of the phylogenetic relationships and the differences in amarillovirus poxins is warranted. On a related note, do the authors have ideas on poxin differences in poxviruses, for example the addition of a schlafen domain.

Thank you for this suggestion. Our maximum-likelihood analysis shows that the relationships between amarillovirus poxins and other groups like poxviruses cannot be resolved with enough certainty to provide a detailed model for how different groups of poxins acquired the ancestral enzyme from RNA viruses. As additional amarillovirus genomes are sequenced, we anticipate that these relationships will become clearer.

We agree that poxin–schlafen fusions in mammalian poxviruses are particularly interesting. Recent work by Hernaez et al., 2020, provides additional interesting analysis of the poxin–schlafen fusion in vivo using ectromelia (mousepox) virus. Their work shows that the poxin domain is critical for subversion of cGAS-STING immunity and for ectromelia virus virulence, and expression of the schlafen domain alone fails to allow viral immune evasion. Further, they show that mammalian schlafen homologs fail to complement a deletion mutant virus lacking the poxin–schlafen fusion. While their study and our work has thus far not elucidated a role for the schlafen domain, its conservation amongst orthopoxviruses as a fusion to poxin strongly suggests a role in restriction of immune signaling. We now include discussion of this newly published work and highlight the potential role of the schlafen domain in immune evasion during infection (see the Discussion). Future biochemical and virology studies will be required to understand the function of the schlafen domain in mammalian poxvirus infection.

Reviewer 2:[…] Overall, this is a very interesting and well written manuscript. The conclusions are well supported by their data and are impactful. Some areas of concern that would improve the manuscript are below:1) Based on the data provided in Figure 2—figure supplement 2, the authors propose that viral enzymes have evolved to have higher binding affinity for cGAMP than host enzymes. However, they are concluding this based on kinetic measurements of only two viral enzymes and only one host enzyme. Could the authors provide kinetic data for a member of clade 4 (viral) and/or a Hymenoptera member of clade 2 (presumably "host") to further support his claim?

We have addressed this point with new experimental data to specifically measure the ability of poxin enzymes to bind 2′3′cGAMP. Building on our biochemical and structural analysis defining the active site residues for VACV, AcNPV, and *T. ni* poxin enzymes, we inactivated each enzyme with a single His>Ala mutation and measured stable poxin–2′3′-cGAMP complex formation using an electrophoretic mobility shift assay. Under these conditions, viral VACV and AcNPV poxin enzymes retain the ability to form a stable complex with 2′3′-cGAMP and exhibit a *K_d_* of ~500–800 nM (see Figure 2—figure supplement 2). In contrast, no stable complex formation is detected for the host *T. ni* poxin enzyme. These results agree with our kinetic analysis demonstrating viral poxins have a lower *K_m_* for 2′3′-cGAMP cleavage and further support a model where host poxins are specifically adapted for immune regulation. However, an important caveat of these experiments is that the alanine substitution in the poxin enzyme active site likely reduces substrate affinity and we note in the figure legend that these experiments therefore underestimate the affinity of poxin for 2′3′-cGAMP.

As our manuscript is focused on detailed characterization of VACV, AcNPV, and *T. ni* poxin enzymes we have elected to not test enzymatic kinetics for further poxin homologs. We have added a line to the text to acknowledge that divergent poxin homologs may exhibit alterative reaction rates (see subsection “Host and viral poxins employ alternative catalytic residues for 2′3′-cGAMP cleavage”). Additionally, we note that the hymenoptera species encoding poxin homologs are species that parasitize lepidopteran insects and lay their eggs within moth and butterfly larvae and it is possible that these poxin enzymes play a similar role to viral poxins in suppressing activation of the lepidopteran immune response.

2) In Figure 6, the authors use a neighbor-joining algorithm to generate phylogenetic trees that they then use to infer deep ancestral relationships of these diverse enzymes. It would be better to use maximum likelihood or Bayesian algorithms to generate these trees, as these approaches are better at resolving deep branches in trees.

Thank you for this suggestion. As noted above in the response to reviewer 1 (see point 1), we have extended our phylogenetic analysis to include a maximum-likelihood tree of diverse poxin sequences with bootstrap analysis as requested.

3) Also in Figure 6, the authors speculate that an ancestral duplication of the viral protease led to the dual-function protease/poxin domains found in amarilloviruses. However, many viral genomes have proteases that are unrelated to each other and are instead distinct folds, etc. Are the N-terminal protease/poxin domain and the other protease in these genomes obviously related to each other? If not, the model should be more agnostic in terms of the exact origin of the protease/poxin domain.

The poxin protease domain is a chymotrypsin-like serine protease fold which is one of the most commonly encoded protease families in (+)ssRNA viruses supporting a possible emergence through internal duplication (Lei and Hilgenfield, 2017; Mann and Sanfaçon, 2019). We have conducted further experiments to assess the activity of amarillovirus poxins, and we now include new experimental data measuring cleavage-site specificity. These results demonstrate that the EQH/ST cleavage site mapped for XCV poxin through Edman degradation sequencing is essential for proteolytic cleavage, and support that the histidine residue at position P1 is particularly important for cleavage site recognition. In addition to including these new data (see Figure 5—figure supplement 2), we agree with the reviewer’s point and have added a question mark in this panel of the model to convey that the exact origin of the ancestral protease is uncertain (see revised Figure 6).

4) The authors state in the Discussion that their data suggest that 2'3'-cGAMP is a predominant ligand in insect immune signaling. Have they shown that these divergent poxins, especially those from amarilloviruses, can exclusively hydrolyze 2'3'-cGAMP?

We have addressed this point with new experimental data to specifically compare the ability of poxin enzymes to cleave 2′3′-cGAMP or the closely related isomer 3′3′-cGAMP. We re-screened representative poxin enzymes from Figure 1 and found that each tested homolog specifically targets 2′3′-cGAMP (see Figure 5—figure supplement 2), with one exception. *Dendrolimus punctatus* cypovirus 22 (DpCV22) poxin shows promiscuous degradation of 3ʹ3ʹ-cGAMP and 2ʹ3ʹ-cGAMP. Our new data indicate that while certain poxin enzymes may have lower substrate specificity, as a whole, poxins broadly maintain a high degree of specificity for 2ʹ3ʹ-cGAMP.